
# Network construction of non-Abelian chiral spin liquids

**Hernan B. Xavier[1], Claudio Chamon,[2] and Rodrigo G. Pereira[1,3]**

**1** Departamento de Física Teórica e Experimental,
Universidade Federal do Rio Grande do Norte, 59072-970 Natal-RN, Brazil
**2** Physics Department, Boston University, Boston, MA, 02215, USA
**3** International Institute of Physics, Universidade Federal do Rio Grande do Norte,
59078-970 Natal-RN, Brazil

## Abstract

We use a network of chiral junctions to construct a family of topological chiral spin liquids in two spatial dimensions. The chiral spin liquid phase harbors $SU(2)_k$ anyons, which stem from the underlying $SU(2)_k$ WZW models that describe the constituent spin chains of the network. The network exhibits quantized spin and thermal Hall conductances. We illustrate our construction by inspecting the topological properties of the $SU(2)_2$ model. We find that this model has emergent Ising anyons, with spinons acting as vortex excitations that bind Majorana zero modes. We also show that the ground state of this network is threefold degenerate on the torus, asserting its non-Abelian character. Our results shed new light on the stability of non-Abelian topological phases in artificial quantum materials.

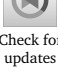

# 1 Introduction

Chiral spin liquids (CSLs) are elusive phases of matter with emergent gauge structures and fractionalized excitations [1,2]. They occur in frustrated quantum magnets with highly entangled ground states that break time-reversal and parity symmetries. A seminal example was put forward by Kalmeyer and Laughlin [3], who constructed a topologically ordered state of spins that bears a strong resemblance to a fractional quantum Hall fluid. Among the similarities, the elementary excitations of CSLs in two dimensions obey anyonic statistics [4]. Of paramount importance to the development of fault-tolerant quantum computing are non-Abelian anyons featuring exotic braiding rules, closely related to the algebraic structure of conformal field theories (CFTs) [5,6]. Examples of non-Abelian CSLs include the Kitaev honeycomb model in a magnetic field [7], the decorated honeycomb model of Yao and Lee [8], and the bosonic Pfaffian wave function proposed by Greiter and Thomale [9].

The low-energy physics of topological CSLs is captured by Chern-Simons theories in 2+1 spacetime dimensions [10]. In addition to anyons, they also feature gapless edge modes [11] and a ground state degeneracy that depends on the topology of space [12]. These properties have been observed in numerical studies of lattice models [13–15]. The edge physics is of great interest to experiments, where one can probe magnetic and thermal responses in the form of quantized Hall conductances [16,17]. Transport measurements can provide a smoking gun to identify CSL phases, as in the recent observation of a quantized thermal Hall conductance for the Kitaev spin liquid candidate material $\alpha$-RuCl$_3$ [18].

In spite of their long history, tractable microscopic models realizing CSL phases are still lacking. In that regard, networks built out of junctions of one-dimensional (1D) electronic systems, such as quantum wires and spin chains, represent a new platform to simulate exotic phases of matter [19–23]. The transport properties of these systems can be characterized by the boundary conditions of the collective modes of charge or spin, which are described by low-energy effective field theories [24,25]. In particular, networks constructed with junctions of spin chains provide a controllable framework to investigate CSLs. Ferraz *et al.* [21] showed how to realize a Kalmeyer-Laughlin state in a honeycomb network of antiferromagnetic spin-$\frac{1}{2}$ Heisenberg chains coupled by three-spin interactions. From the perspective of the effective field theory, the starting point is a junction of SU(2)$_1$ Wess-Zumino-Witten (WZW) models with boundary interactions tuned to a chiral fixed point [26,27]. In this approach, one can directly access the topological properties of the CSL without invoking a renormalization group flow towards a strong-coupling fixed point as in standard coupled-wire constructions [28–33]. While the Kalmeyer-Laughlin state studied in Ref. [21] is an example of Abelian topological phase, the recent finding of a chiral fixed point of critical spin-1 chains described by the SU(2)$_2$ WZW model [34] opens the way for generalizations with non-Abelian anyons.

In this paper, we extend the network construction to non-Abelian CSLs. Postulating the existence of chiral fixed points in junctions of SU(2)$_k$ WZW models, we assemble a honeycomb network that realizes gapped CSL phases, see Fig. 1. The network harbors elementary spin-$j$ excitations of all spins from $j = \frac{1}{2}$ to $j = \frac{k}{2}$, with energy gaps determined by the scaling dimensions of the primary fields of the underlying WZW models. The CSL exhibits quantized spin and thermal Hall conductances that depend on the level $k$ of the WZW model. We show that perturbations to the chiral fixed point control the mobility of the elementary excitations, and the CSL phase is stable over a finite range of couplings for which all excitations remain gapped. We test the topological properties of our construction by studying the SU(2)$_2$ model as a concrete example. This network model has a factorized spectrum akin to Kitaev's non-Abelian CSL, with spin-$\frac{1}{2}$ excitations (spinons) acting as $\mathbb{Z}_2$ vortices that bind Majorana zero modes. We also show that the ground state of the SU(2)$_2$ CSL on the torus is threefold degenerate,

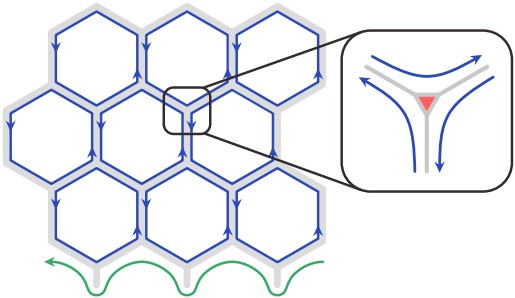

Figure 1: Honeycomb network of critical spin chains realizing a gapped chiral spin liquid phase. Blue arrows indicate the direction of spin currents in the bulk. Edge modes are represented in green. The inset shows how the low-energy modes are rerouted in the anticlockwise direction at a junction tuned to the chiral fixed point.

as a consequence of the blocking mechanism that demonstrates the non-Abelian character of this phase.

This paper is organized as follows. We start in Sec. 2 with a broad view of the network construction of $SU(2)_k$ CSLs. After presenting our construction scheme, we verify that the spectrum contains $SU(2)_k$ anyons. To further characterize our CSLs, we compute their spin and thermal Hall conductances from the edge modes in a strip geometry. We conclude this overview with a discussion about the stability of the CSL phases by looking into the effects of boundary perturbations allowed by symmetry. Section 3 provides an analysis of the $SU(2)_2$ model, which has emergent Ising anyons and a threefold topological degeneracy on the torus. Finally, we draw our conclusions and point out some future directions in Sec. 4. We also include an Appendix where we detail the computation of some commutators for string operators on the network.

## 2 Network construction for $SU(2)_k$ models

Quantum spin circulators [26, 27] can be viewed as building blocks of networks that harbor CSL phases. In this section, we revisit the major plot points of this construction, extending the results of Ref. [21] to higher-level $SU(2)_k$ WZW models, which describe critical points in the phase diagrams of isotropic spin-$S$ chains [35–40]. Motivated by the recent report of a chiral fixed point in the junction of $SU(2)_2$ models [34], we start by postulating the existence of chiral fixed points for general values of $k$ and examine the properties of the corresponding CSLs. A posteriori, the analysis of perturbations to the putative chiral fixed points allows us to assess the stability of these non-Abelian phases.

### 2.1 Low-energy spectrum

Consider a honeycomb network constructed by coupling together a large number of finite spin chains that, when isolated, are described by independent $SU(2)_k$ WZW models. The bulk Hamiltonian of a single spin chain is written in Sugawara form as [41]

$$H_c = \frac{2\pi v}{k+2} \int_0^\ell dx \big( \mathbf{J}_{L,c}^2 + \mathbf{J}_{R,c}^2 \big), \tag{1}$$

where $v$ is the spin velocity, $\ell$ is the length of the chain, and $k$ is the level of the WZW model. The operators $\mathbf{J}_{L,c}$ and $\mathbf{J}_{R,c}$ are the left- and right-moving currents that propagate on chain c.

We then assume that we can tune the boundary interactions in a microscopic model for the junctions to a chiral fixed point [26, 27, 34]. The key property of this fixed point is that at each junction the incoming currents are perfectly transmitted to the next chain in rotation, as in an ideal circulator; see the inset in Fig. 1. As a result, the currents circulate in loops. When we impose chiral boundary conditions at the junctions, the set of currents associated with the spin chains can be locally mapped to plaquette spin currents $\mathbf{J}_p$,

$$(\mathbf{J}_{\text{L,c}}, \mathbf{J}_{\text{R,c}}) \mapsto (\mathbf{J}_p, \mathbf{J}_{p'}),\tag{2}$$

where $p$ and $p'$ are plaquettes sharing chain c. In terms of the plaquette currents, the effective low-energy Hamiltonian of the network assumes the form

$$H = \sum_p H_p, \qquad H_p = \frac{2\pi v}{k+2} \int_0^L dx\, \mathbf{J}_p^2,\tag{3}$$

where $L = 6\ell$ is the distance traveled by a chiral mode around a plaquette and $\mathbf{J}_p$ is subjected to periodic boundary conditions

$$\mathbf{J}_p(x+L) = \mathbf{J}_p(x).\tag{4}$$

Since the currents are confined to the plaquettes, the network has a gapped energy spectrum in the bulk. To diagonalize the Hamiltonian in Eq. (3), we proceed to momentum space. Using translation invariance, we expand the spin current in Fourier modes as

$$J_p^a(x) = \frac{1}{L} \sum_{n=-\infty}^{\infty} J_{p,n}^a e^{i2\pi nx/L} \quad (n \in \mathbb{Z}),\tag{5}$$

where $a = \{x, y, z\}$ are the spin components. The modes satisfy the commutation relations of the Kac-Moody algebra, $[J_{pn}^a, J_{qm}^b] = i\epsilon^{abc}\delta_{pq}J_{n+m}^c + \frac{1}{2}kn\delta^{ab}\delta_{pq}\delta_{n+m,0}$. Substituting the mode expansion into the Hamiltonian, we obtain

$$H = \sum_p \frac{2\pi v}{L} \frac{1}{k+2} \Big( J_{p,0}^a J_{p,0}^a + 2 \sum_{n>0} J_{p,-n}^a J_{p,n}^a \Big),\tag{6}$$

where we drop an unimportant additive constant and sum over repeated spin indices. To label the eigenstates of the Hamiltonian, we use the quantum numbers of total spin and total spin-$z$ projection operators, $\mathbf{S}_{\text{tot}}^2$ and $S_{\text{tot}}^z$. These quantum numbers are fixed by the eigenvalues of the plaquette zero-mode operators, in the form

$$\mathbf{S}_{\text{tot}} \equiv \sum_p \mathbf{S}_p, \qquad \mathbf{S}_p = \int_0^L dx\, \mathbf{J}_p(x) = \mathbf{J}_{p,0}.\tag{7}$$

Note that the effective theory for the chiral fixed point is endowed with an enlarged symmetry, as the Hamiltonian in Eq. (3) commutes with each $\mathbf{S}_p$ individually. However, the quantum numbers of $\mathbf{S}_{\text{tot}}^2$ and $S_{\text{tot}}^z$ impose global constraints on the network spectrum. If we assume that the total number of lattice sites in the network is even, the allowed values for $S_{\text{tot}}^z$ are integers, which precludes the existence of single spin-$\frac{1}{2}$ excitations.

The ground state of the system is a spin singlet defined by the vacuum condition

$$J_{p,n}^a |\Omega\rangle = 0,\tag{8}$$

for all plaquettes $p$, all spin components $a$, and non-negative integers $n$. The low-energy Hilbert space is generated by acting on the vacuum with operators of the chiral SU(2)$_k$ WZW

models associated with the plaquettes [41, 42]. The first excited state is highly degenerate and corresponds to a pair of elementary spin-$\frac{1}{2}$ excitations, known as spinons, with energy

$$E_1 = \frac{\pi v}{L} \frac{3}{k+2} \,. \tag{9}$$

In fact, the network carries elementary excitations with spin $j = \frac{1}{2}, 1, \cdots, \frac{k}{2}$. They are created in pairs by the action of local operators in the spin chains, which can be written in terms of the primary fields $\Phi_c^{(j)}$ of the SU(2)$_k$ WZW model. Crucially, in our construction the left- and right-moving parts of $\Phi_c^{(j)}$ act on different, neighboring plaquettes of the network. The energy of a pair of elementary spin-$j$ excitations is

$$E_{2j} = \frac{4\pi v}{L} h_j \,, \qquad h_j = \frac{j(j+1)}{k+2} \,, \tag{10}$$

where $2h_j$ is the scaling dimension of $\Phi_c^{(j)}$. Note that the excited states are degenerate with respect to the plaquettes in which the spin-$j$ excitations are located. Although a spin-$j$ pair can only be created at neighboring plaquettes, one can move them apart by a series of local operations without energy cost. Finally, each elementary spin-$j$ excitation has a tower of descendant states, which can be obtained by applying ladder operators $J_{p,-n}^a$ with $n \geq 1$. For high values of $k$, the first excited state with a large spin $j$ may lie above the first descendant in the tower of the identity operator, which corresponds to $j = 0$.

## 2.2 Edge modes and transport properties

The edge theory of the network is governed by a chiral SU(2)$_k$ WZW model. The gapless edge modes can be directly visualized in real space by tracking the path of the currents reflected at the open ends of the spin chains at the boundary, see Fig. 1. The edge modes carry spin and energy, contributing to transport properties of the CSL phase [11, 16, 17]. Here we show that the network exhibits a quantized response to gradients of magnetic fields and temperature. In a strip geometry, we can treat the edge modes as spatially separated chiral currents $\mathbf{J}_{\text{upper}}$ and $\mathbf{J}_{\text{lower}}$, with Hamiltonian

$$H_{\text{edge}} = \frac{2\pi v}{k+2} \int dx \left( \mathbf{J}_{\text{upper}}^2 + \mathbf{J}_{\text{lower}}^2 \right), \tag{11}$$

where $x \in \mathbb{R}$ runs along the edge. Figure 2 shows a strip whose width is much larger than the chain length $\ell$. At large length scales, the edge modes propagate approximately in the direction indicated by the $x$ axis. The detailed geometry of the boundary can be absorbed into a rescaling of the velocity for the edge modes, which does not affect the quantized transport coefficients.

To evaluate the spin Hall response, we apply a magnetic field $h$ at the upper edge of the strip. This perturbation imposes a spin voltage drop along the transverse direction of the strip. For small magnetic fields, $h \ll v/L$, we can neglect the bulk contribution and concentrate on

$$\delta H_{\text{edge}} = h \int dx \, J_{\text{upper}}^z \,. \tag{12}$$

Due to the magnetic field difference, we observe a nonzero longitudinal spin current $I_s = v \langle J_{\text{upper}}^z - J_{\text{lower}}^z \rangle_h$. From standard linear response theory, we find that the spin Hall conductance is given by

$$G_{xy} = - \lim_{\omega \to 0^+} v \int dx \int d\tau \, e^{i\omega\tau} \langle J_{\text{upper}}^z(\tau, x) J_{\text{upper}}^z(0) \rangle \,, \tag{13}$$

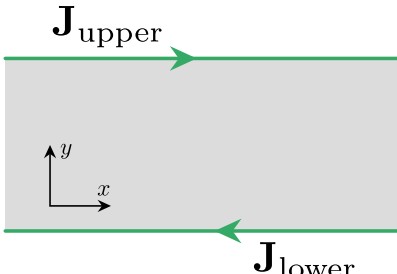

Figure 2: Effective strip geometry. Hall responses are evaluated by subjecting the system to gradients of magnetic fields and temperature along the transverse direction. Green arrows represent the direction of propagation of the chiral edge modes. The gapped bulk is depicted in gray.

being closely related to the magnetic susceptibility of a chiral WZW model [43]. If we then use the current correlator

$$\langle J^a_{\text{upper}}(\tau, x) J^b_{\text{upper}}(0)\rangle = \frac{k}{8\pi^2} \frac{\delta^{ab}}{(v\tau - ix)^2}\,, \tag{14}$$

and perform the integrations, we arrive at

$$G_{xy} = \frac{k}{4\pi}\,. \tag{15}$$

Thus, the network exhibits a quantized spin Hall conductance $G_{xy}/G_0 = k/2$, where $G_0 = 1/2\pi$ is the spin conductance quantum. Note that, generally speaking, $k/2$ coincides with the spin $S$ of critical spin chain models whose low-energy physics is described by an $SU(2)_{k=2S}$ WZW model [35]. In particular, our formula correctly reproduces $G_{xy} = 1/4\pi$ for $k = 1$, as predicted for the network of spin-$\frac{1}{2}$ Heisenberg chains [21].

We now consider the thermal Hall response of the system. For this part we follow closely the calculations of Cappelli *et al.* [17], who showed that the thermal Hall conductance of a quantum Hall state is directly proportional to the central charge $c$ of the chiral CFT on its edge. We assume that the upper edge is held at finite temperature $T$, creating a temperature bias along the transverse direction of the strip. The induced thermal current is determined by the expectation value of the energy-momentum tensors:

$$I_Q = \frac{v^2}{2\pi} \langle \mathcal{T}_{\text{upper}} - \mathcal{T}_{\text{lower}}\rangle_{\Delta T}\,. \tag{16}$$

As the lower edge is kept at zero temperature, the expectation value of its energy-momentum tensor vanishes, $\langle \mathcal{T}_{\text{lower}}\rangle_0 = 0$. Let us then focus on the contribution from the upper edge. We extend our CFT to finite temperature using the conformal mapping $w \to z(w) = e^{i2\pi w/\beta}$, where $\beta = 1/T$ is the inverse temperature, and $z$ and $w$ represent the coordinates on the cylinder and on the plane, respectively [42]. When we perform this conformal transformation, the energy-momentum tensor acquires the nonzero expectation value:

$$\langle \mathcal{T}_{\text{upper}}(z)\rangle_T = \left(\frac{dz}{dw}\right)^2 \langle \mathcal{T}_{\text{upper}}(w)\rangle_0 + \frac{c}{12}\{z; w\} = \frac{\pi^2}{6}\frac{cT^2}{v^2}\,. \tag{17}$$

Here, $\{z; w\}$ stands for the Schwarz derivative, defined as $\{z; w\} = \frac{z'''}{z'} - \frac{3}{2}(\frac{z''}{z'})^2$ [44]. From Eqs. (16) and (17), we see that the induced thermal current is equal to $I_Q = \frac{\pi}{12}cT^2$. This implies that the network has a quantized thermal Hall conductance

$$K_{xy} = \frac{\partial I_Q}{\partial T} = \frac{\pi}{6}cT = \frac{\pi}{2}\frac{k}{k+2}T\,. \tag{18}$$

In the last equality we used the expression for the central charge of the $SU(2)_k$ WZW model, $c = 3k/(k+2)$ [41].

## 2.3 Perturbations and stability

Reaching the chiral fixed point of junctions of $SU(2)_k$ WZW models requires fine tuning microscopic boundary interactions [26, 27, 34]. Once the model parameters deviate from this special point, boundary perturbations appear at the junctions of the network. These perturbations couple the low-energy modes in neighboring plaquettes, allowing the elementary excitations to become mobile and lower their energy. However, the stability of the CSL phase is guaranteed by the excitation gap for a network composed of chains with finite length.

For a single junction, the leading boundary operators allowed by symmetry that perturb the chiral fixed point are

$$\delta H_Y = \sum_{j=1/2}^{k/2} \lambda_{2j} \sum_{c=1}^{3} \text{tr}\, \Phi_c^{(j)}(0), \tag{19}$$

where $\lambda_{2j}$ are coupling constants. Here we take the trace of the primary matrix fields to obtain SU(2)-invariant operators and we sum over the three chains $c = 1, 2, 3$ that make up the junction, preserving the $\mathbb{Z}_3$ cyclic permutation symmetry. The boundary operators are placed at the junction point $x = 0$. Recall that $\text{tr}\, \Phi_c^{(j)}$ has scaling dimension $2h_j$. Thus, for large values of $k$, $\delta H_Y$ may include not only relevant, but also marginal and irrelevant boundary perturbations. We neglect boundary operators that are written in terms of the currents, such as $J_{L,c}^2(0)$, because they are always irrelevant.

Let $\mathcal{N}_k$ denote the number of relevant and marginal boundary operators, for which $2h_j \leq 1$. From Eq. (10), we have $\mathcal{N}_k = \lfloor \sqrt{2k+5} - 1 \rfloor$. To reach the chiral fixed point, we need to tune a set of parameters in the microscopic model, $\{g_{2j}\}$ with $j = \frac{1}{2}, \cdots, \frac{1}{2}\mathcal{N}_k$, to a special point $\{g_{2j}^\star\}$ in the $\mathcal{N}_k$-dimensional boundary phase diagram so that $\lambda_{2j} = 0$. In the simplest example, $k = 1$, it suffices to tune a single parameter, namely the strength of the three-spin interaction at the boundary [26, 27]. The model with $k = 2$ is special because it contains one relevant and one marginal operator [34].

Moving on to the 2D network, the boundary perturbations can be expressed as

$$\delta H = \sum_{j=1/2}^{k/2} \lambda_{2j} \sum_{c} \sum_{r=1}^{2} \text{tr}\, \Phi_c^{(j)}(x_r). \tag{20}$$

Here, the summation runs over all chains $c$. Since every spin chain terminates at two junctions, we use the positions $x_1 = 0^+$ and $x_2 = \ell^-$ to parametrize the two ends of the chain.

In general, these boundary operators break the integrability of the model. We expect that their leading effect is to lift the extensive degeneracy of the excited states discussed in Sec. 2.1. We can unveil this physics by applying degenerate perturbation theory. Let $|\phi_p^{(j)}\rangle$ be a one-particle state corresponding to an elementary spin-$j$ excitation at plaquette $p$. Although such a state is unphysical, we may still use it as an approximation to the situation where a pair of excitations is taken very far apart. To first order, the perturbation couples one-particle states located in neighboring plaquettes, generating an effective hopping amplitude given by the matrix element

$$t_{2j} = \langle \phi_p^{(j)} | \delta H | \phi_{p'}^{(j)} \rangle. \tag{21}$$

We can use the properties of the WZW model to extract the scaling behavior of $t_{2j}$. First, we recall that the primary fields act on chiral modes that belong to adjacent plaquettes. This means that we can fractionalize $\Phi_c^{(j)}$ into

$$\Phi_c^{(j)} \mapsto \phi_p^{(j)} \times \phi_{p'}^{(j)}, \tag{22}$$

where $\phi_p^{(j)}$ denotes the chiral primary field with spin $j$ at plaquette $p$. Second, from the fusion rules of the chiral WZW model [44]

$$\phi_p^{(j)} \times \phi_{p'}^{(j')} = \sum_{l=|j-j'|}^{\min(j+j',k-j-j')} \delta_{pp'} \phi_{p'}^{(l)}, \tag{23}$$

we deduce that the action of $\phi_p^{(j)}$ onto the states $|\phi_{p'}^{(j')}\rangle$ only returns the ground state for $j = j'$:

$$\phi_p^{(j)} |\phi_{p'}^{(j')}\rangle = \delta_{pp'} \delta_{jj'} |\Omega\rangle + \cdots \tag{24}$$

This means that the problem factorizes, with spin-$j$ excitations only being scattered by the coupling $\lambda_{2j}$. Hence, $t_{2j}$ obeys the scaling law

$$t_{2j} \propto \lambda_{2j} \ell^{-2h_j}. \tag{25}$$

As a result, to first order in perturbation theory the first excited state in the spin-$j$ sector splits into a continuum with bandwidth proportional to $|t_{2j}|$. A quantum phase transition occurs once the perturbation is large enough to close the energy gap, i.e. for $|t_{2j}| \sim v/\ell$. When $\lambda_{2j}$ is a relevant coupling constant, the region of stability of the gapped CSL phase shrinks with increasing chain length, reflecting the instability of the chiral fixed point in a single junction with $\ell \to \infty$ [21]. Using $\lambda_{2j} \propto g_{2j} - g_{2j}^\star$, we find that the CSL is stable for

$$\frac{1}{\ell} \gtrsim |g_{2j} - g_{2j}^\star|^{1/(1-2h_j)}. \tag{26}$$

Remarkably, the boundary perturbations in our network model have an effect similar to that of exchange interactions that break the integrability of the Kitaev honeycomb model and generate a mobility for visons in a Kitaev spin liquid [45, 46].

# 3 Ising topological order in the network of SU(2)$_2$ models

We now consider the SU(2)$_2$ CSL as a particular example of our construction. This model may be realized in a network of critical spin-1 chains [34]. Our main goal here is to verify the topological properties of this CSL phase, asserting its non-Abelian character.

## 3.1 Majorana formulation

Let us begin by noting that SU(2)$_2$ anyons can be mapped onto Ising anyons. The chiral SU(2)$_2$ WZW model contains three primary fields, including the identity $\phi^{(0)} = \mathbb{1}$, with fusion rules [see Eq. (23)]

$$\phi^{(\frac{1}{2})} \times \phi^{(\frac{1}{2})} = \mathbb{1} + \phi^{(1)}, \qquad \phi^{(\frac{1}{2})} \times \phi^{(1)} = \phi^{(\frac{1}{2})}, \qquad \phi^{(1)} \times \phi^{(1)} = \mathbb{1}. \tag{27}$$

These fusion rules can be identified with those of the Ising CFT with scaling fields $\mathbb{1}$, $\sigma$, and $\xi$ [44],

$$\sigma \times \sigma = \mathbb{1} + \xi, \qquad \sigma \times \xi = \sigma, \qquad \xi \times \xi = \mathbb{1}, \tag{28}$$

if we map $\phi^{(\frac{1}{2})} \mapsto \sigma$ and $\phi^{(1)} \mapsto \xi$. On the other hand, it is important to keep in mind that the energy levels of the SU(2)$_2$ theory feature an additional spin degeneracy associated with the SU(2) spin-rotation symmetry of the Hamiltonian.

In fact, the $SU(2)_2$ WZW model can be expressed as a theory of three Ising models [41, 47–49]. In this formulation, we can write the plaquette Hamiltonian in terms of three chiral Majorana fermions

$$H_p = -\frac{iv}{2} \int_0^L dx\, \xi_p^a \partial_x \xi_p^a, \tag{29}$$

where we sum over repeated spin indices $a = x, y, z$. The fermions obey standard anticommutation relations, $\{\xi_p^a(x), \xi_q^b(y)\} = \delta^{ab}\delta_{pq}\delta(x-y)$. The spin currents are given by

$$J_p^a(x) = -\frac{i}{2}\epsilon^{abc}\xi_p^b(x)\xi_p^c(x), \tag{30}$$

for all spin components $a$ in the plaquette $p$. The periodic boundary conditions for the currents can be satisfied by imposing

$$\xi_p^a(x+L) = \pm\xi_p^a(x). \tag{31}$$

These two types of boundary conditions give two independent sectors of the theory in the plaquette. Antiperiodic boundary conditions define the Neveu-Schwarz (NS) sector, while periodic boundary conditions specify the Ramond (R) sector. To label these sectors, we introduce the $\mathbb{Z}_2$ variable $w_p$ defined such that

$$w_p = \begin{cases} +1, & \text{antiperiodic BCs} \quad \text{(NS sector)}, \\ -1, & \text{periodic BCs} \qquad \text{(R sector)}. \end{cases} \tag{32}$$

The full set of $w_p$ defines a static $\mathbb{Z}_2$ flux configuration. The $\mathbb{Z}_2$ flux can be determined by assigning signs in the chiral boundary conditions for the Majorana fermions at each junction around a plaquette, which is equivalent to fixing a gauge in the representation of the local operators [34]. In analogy with the solution of the Kitaev honeycomb model [7], we write the eigenstates of $H = \sum_p H_p$ in the factorized form

$$|\Psi\rangle = |\mathcal{M}_\mathcal{G}\rangle|\mathcal{G}\rangle, \tag{33}$$

where $|\mathcal{M}_\mathcal{G}\rangle$ is a many-body eigenstate of Majorana fermions in the background of the $\mathbb{Z}_2$-field configuration specified by $|\mathcal{G}\rangle$.

To find the spectrum for given boundary conditions, we go to momentum space. The mode expansion for the chiral fermions has the general form

$$\xi_p^a(x) = \frac{1}{\sqrt{L}} \sum_{k=-\infty}^{\infty} \xi_{p,k}^a e^{i2\pi kx/L}, \tag{34}$$

where $k$ is half-integer in the case of antiperiodic boundary conditions, and integer otherwise. When we substitute the mode expansion into the plaquette Hamiltonian, we have to distinguish between the NS and R sectors:

$$\begin{aligned} H_p &= \frac{2\pi v}{L}\sum_{k>0} k\xi_{p,-k}^a\xi_{p,k}^a - \frac{2\pi v}{L}\frac{1}{16}, \qquad (k \in \mathbb{Z} + \tfrac{1}{2}), \\ H_p &= \frac{2\pi v}{L}\sum_{k>0} k\xi_{p,-k}^a\xi_{p,k}^a + \frac{2\pi v}{L}\frac{1}{8}, \qquad (k \in \mathbb{Z}). \end{aligned} \tag{35}$$

The additive constants come from the normal ordering of the mode operators and can be obtained from the regularization prescription given by the Riemann zeta function [44].

We see that the ground state of $H_p$ is the NS vacuum, while the R vacuum is the first excited state. Thus, changing the $\mathbb{Z}_2$ flux from $w_p = 1$ to $w_p = -1$ costs energy, and we can associate the R vacuum to a vortex excitation. The corresponding single-vortex gap is

$$E_{\mathrm{v}} = \frac{2\pi v}{L}\frac{3}{16}\,. \tag{36}$$

Moreover, the ground state in the R sector is degenerate due to the zero-mode operators $\xi^a_{p,0}$ which do not enter the Hamiltonian. These operators commute with $H_p$ and satisfy $(\xi^a_{p,0})^2 = \frac{1}{2}$. For each plaquette there are three zero modes, whose degeneracy is protected by the SU(2) symmetry. We can combine the Majorana zero modes to form a fundamental representation of the SU(2) algebra (omitting the plaquette index):

$$s^x = -i\xi^y_0\xi^z_0\,, \qquad s^y = -i\xi^z_0\xi^x_0\,, \qquad s^z = -i\xi^x_0\xi^y_0\,, \tag{37}$$

with $[s^a, s^b] = i\epsilon^{abc}s^c$ and $\mathbf{s}^2 = 3/4$. As a result, the vortex corresponds to an elementary spin-$\frac{1}{2}$ excitation. In other words, the spinon of the $\mathrm{SU}(2)_2$ model binds Majorana zero modes. This conclusion is consistent with the fusion rules in Eq. (27), which tell us that the spinon has two fusion channels and should be responsible for the non-Abelian character of the CSL.

We are now in a position to recover the spectrum of the network. The ground state corresponds to the vacuum state on the vortex-free configuration, i.e. $w_p = 1$ for all plaquettes $p$. The first excited state compatible with the global constraints is a two-vortex configuration, with energy

$$E_{2\mathrm{v}} = \frac{2\pi v}{L}\frac{3}{8}\,. \tag{38}$$

The elementary spin-1 excitations are represented by the Majorana fermions. The lowest-energy state in this subspace that respects global fermion parity is a two-fermion excitation $\xi^a_{p,-1/2}\xi^b_{q,-1/2}|\Omega\rangle$, with energy

$$E_{2\mathrm{f}} = \frac{2\pi v}{L}\,. \tag{39}$$

Note that these excitation energies are compatible with the general formula in Eq. (10).

Let us dig a bit deeper and see how these excitations are created by the action of local operators. We recall that the $\mathrm{SU}(2)_2$ WZW model that describes a spin chain has two nontrivial primary fields, $\Phi^{(\frac{1}{2})}_{\mathrm{c}}$ and $\Phi^{(1)}_{\mathrm{c}}$. We first examine the spin-1 operator, whose components can be expressed as Majorana bilinears [41]. We write the spin-1 matrix field of a given chain in terms of the chiral modes in the network as

$$[\Phi^{(1)}_{\mathrm{c}}]^{ab} = i\xi^a_{\mathrm{L,c}}\xi^b_{\mathrm{R,c}} \mapsto i\xi^a_p\xi^b_{p'}\,, \tag{40}$$

where $p$ and $p'$ are the two plaquettes sharing chain c. This mapping implies that $\Phi^{(1)}_{\mathrm{c}}$ fractionalizes into a pair of Majorana fermions at adjacent plaquettes. Working in the basis of fermionic ladder operators:

$$(\xi^x, \xi^y, \xi^z) \to (\xi^+, \xi^-, \xi)\,, \tag{41}$$

with $\xi^{\pm} = \xi^x \pm i\xi^y$ and $\xi = \xi^z$, we can easily verify that these fermions represent spin-1 excitations:

$$[S^z_{\mathrm{tot}}, \xi_p(x)] = 0\,, \qquad [S^z_{\mathrm{tot}}, \xi^{\pm}_p(x)] = \pm\xi^{\pm}_p(x)\,. \tag{42}$$

Note that we have dropped the upper index for the third Majorana fermion in the ladder representation to lighten the notation. Importantly, these excitations are deconfined on the network, meaning that we can move them without energy cost. For example, if we start with

the state $\xi^+_{p_2}\xi^-_{p_1}|\Omega\rangle$, we can apply a series of local operations $\xi^+_{p_n}\xi^-_{p_{n-1}}\cdots\xi^+_{p_3}\xi^-_{p_2}$ to move the Majorana from plaquette $p_2$ to $p_n$. For elementary Majorana excitations, the final state

$$\xi^+_{p_n,-1/2}\xi^-_{p_{n-1},-1/2}\cdots\xi^+_{p_2,-1/2}\xi^-_{p_1,-1/2}|\Omega\rangle\,,\tag{43}$$

is also an eigenstate of the network with the same energy given in Eq. (39). While we may use this property to send one Majorana to infinity and talk about single Majorana states, physical states of the network always contain an even number of Majorana fermions.

It is also convenient to bosonize the complex fermions in the basis of fermionic ladder operators. We write

$$\xi^\pm_p(x)\sim\frac{1}{\sqrt{\pi}}\exp[\pm 2i\phi_p(x)]\,.\tag{44}$$

To ensure the anticommutation of fermions, we impose the equal-time algebra for the chiral bosons

$$[\phi_p(x),\phi_p(y)]=i\frac{\pi}{4}\,\mathrm{sgn}(x-y)\,,\tag{45}$$

where $\mathrm{sgn}(x)$ is the sign function defined so that $\mathrm{sgn}(0)=0$. The plaquette magnetization $S^z_p$, see Eq. (7), has a simple representation in terms of the chiral boson:

$$S^z_p=\frac{1}{2}\int_0^L dx\;{:}\xi^+_p\xi^-_p{:}=\frac{1}{\pi}\int_0^L dx\,\partial_x\phi_p\,,\tag{46}$$

and hence $S^z_p=(1/\pi)\big[\phi_p(L)-\phi_p(0)\big]$. Since the total spin of the network is an integer, we have the constraint

$$1=e^{i2\pi S^z_{\mathrm{tot}}}=\prod_p e^{i2[\phi_p(L)-\phi_p(0)]}\,.\tag{47}$$

Next, we consider the spin-$\frac{1}{2}$ matrix field. The latter appears in the staggered magnetization of a spin chain, given by $\mathbf{n}_c\propto\mathrm{tr}\,\tau\Phi^{(\frac{1}{2})}_c$, where $\tau^a$ with $a=x,y,z$ are Pauli matrices [35]. The $n^\pm_c$ components obey the commutation relations with the chiral currents

$$[J^z_{\mathrm{L/R,c}}(x),n^\pm_c(y)]=\pm\frac{1}{2}\delta(x-y)n^\pm_c(y)\,.\tag{48}$$

Given the mapping in Eq. (2), this means that $n^\pm_c$ creates two spin-$\frac{1}{2}$ excitations at adjacent plaquettes. In practice, we can say the action of $n^\pm_c$ fractionalizes into

$$n^\pm_c\mapsto u^\pm_p u^\pm_{p'}\,,\tag{49}$$

where $u^\pm$ are chiral twist operators. In the notation where $\xi^\pm$ are bosonized, the SU(2)$_2$ twist operator takes the form

$$u^\pm_p\sim e^{\pm i\phi_p}\sigma_p\,,\tag{50}$$

where $\sigma_p$ is the twist operator for the Majorana $\xi_p$ [50].

The twist operator changes the boundary conditions of the fermions on the plaquettes from antiperiodic to periodic and vice versa. To verify that, we first note that the action of the vertex operator $e^{\pm i\phi_p}$ on the boson $\phi_p$ gives

$$e^{\mp i\phi_p(y)}\phi_p(x)e^{\pm i\phi_p(y)}=\phi_p(x)\pm\frac{\pi}{4}\,\mathrm{sgn}(x-y)\,.\tag{51}$$

Thus, $e^{\pm i\phi}$ creates a kink in the bosonic field configuration. When we go around the plaquette, we gather the phase shift

$$\phi_p(L)-\phi_p(0)\to\phi_p(L)-\phi_p(0)\pm\frac{\pi}{2}\,.\tag{52}$$

This phase shift reverses the boundary conditions of $\xi_p^\pm$, see Eq. (44), creating one spinon with $S_p^z = \pm\frac{1}{2}$.

The next step is to verify how $\sigma_p$ changes the boundary conditions of $\xi_p$. Their OPE has the form

$$\xi_p(z)\sigma_p(0) \sim z^{-1/2}\sigma_p(0) + \cdots \tag{53}$$

Due to the branch cut introduced by the twist operator, for $z = e^{i\theta}$, we get different signs whether $\theta = 0$ or $\theta = 2\pi$. This is equivalent to imposing the equal-time relations (dropping the plaquette index):

$$\xi(x)\sigma(y) = \begin{cases} +\sigma(y)\xi(x), & x < y, \\ -\sigma(y)\xi(x), & x > y. \end{cases} \tag{54}$$

The sign choice is arbitrary and depends on where we place the branch cut. (We can also think the other choice is implemented by the dual twist field $\mu$). Independently of our choice, we can show that $\sigma$ implements periodic boundary conditions when acting on the NS vacuum:

$$\xi(L)\sigma(x)|0\rangle_{\mathrm{NS}} = -\sigma(x)\xi(L)|0\rangle_{\mathrm{NS}} = \sigma(x)\xi(0)|0\rangle_{\mathrm{NS}} = \xi(0)\sigma(x)|0\rangle_{\mathrm{NS}}, \tag{55}$$

where $0 < x < L$ is an arbitrary position at the plaquette. In sum, the action of $u^\pm$ upon the NS vacuum yields the R vacuum (plus descendants),

$$u^\pm(x)|0\rangle_{\mathrm{NS}} \sim |\pm\rangle_{\mathrm{R}} + \cdots \tag{56}$$

To move a pair of spinons, we must apply an ordered product of twist operators. The multiplication rules of the vertex operators, together with the fusion rules of the Ising model, yield the mutual OPEs:

$$u^\pm(z)u^\pm(w) \sim (z-w)^{1/8}\xi^\pm(w) + \cdots,$$
$$u^\pm(z)u^\mp(w) \sim \frac{1}{(z-w)^{3/8}} + (z-w)^{1/8}\xi(w) + \cdots \tag{57}$$

Here we omit the structure constants and the ellipsis stands for less relevant terms. Note that our OPEs are consistent with the fusion rules of the chiral $SU(2)_2$ WZW model in Eq. (27). Since the product $u^\pm \times u^\mp$ has two fusion channels, we define the two-spinon excitation as

$$u_{p_n}^+ \Big( \mathcal{P}_{\mathbb{1}} u_{p_{n-1}}^- u_{p_{n-1}}^+ \cdots \mathcal{P}_{\mathbb{1}} u_{p_2}^- u_{p_2}^+ \mathcal{P}_{\mathbb{1}} \Big) u_{p_1}^- |\Omega\rangle, \tag{58}$$

where $\mathcal{P}_{\mathbb{1}}$ is the projection onto the identity channel. The factors of $\mathcal{P}_{\mathbb{1}}$ guarantee that no fermions are created along the string in the process of separating the spinons.

## 3.2 Topological degeneracy on the torus

Topological phases exhibit a ground state degeneracy that depends on the topology of space. Let us assure the topological order of the CSL phase by computing its topological degeneracy on the torus. This can be done by inspecting the algebra of string operators that implement the transport of quasiparticles around the noncontractible directions of the torus [50, 51].

We first define $T_x$ as the operator that transports a spin-1 quasiparticle along the horizontal direction of the network. We write $T_x$ as a product of ladder operators $\xi^\pm$ in the form

$$T_x \sim \prod_{p \in \mathcal{W}_x} \xi_p^+(\ell + \epsilon)\xi_p^-(-\ell - \epsilon), \tag{59}$$

where the product is taken over all plaquettes $p$ crossed by the closed string $\mathcal{W}_x$, which winds around the torus in direction depicted in Fig. 3. Note that we have a free parameter $0 < \epsilon < \ell$

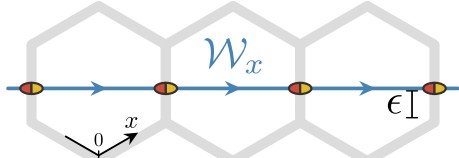

Figure 3: String operator that realizes the transport of quasiparticles along the horizontal direction. Here the periodic plaquette coordinate $x \in [-3\ell, 3\ell]$ is measured from the bottom vertex of the hexagon in the anticlockwise direction. The free parameter $0 < \epsilon < \ell$ controls the position where we cut the plaquettes.

that determines the position where the string cuts the plaquettes. We can also use bosonization, see Eq. (44), to define $T_x$ as

$$T_x \sim \prod_{p \in \mathcal{W}_x} \exp\left[ 2i \int_{-\ell-\epsilon}^{\ell+\epsilon} dx\, \partial_x \phi_p(x) \right]. \tag{60}$$

This operator has three key properties we would like to highlight. First, as one can readily verify, $T_x$ is unitary, i.e. $T_x^\dagger T_x \sim \mathbb{1}$. Second, it gives information about the global boundary conditions on the torus. Under a proper normalization, $T_x$ has eigenvalues $\pm 1$. This follows from the fact that when we annihilate the Majorana pair, after completing one turn around the torus, the transported Majorana picks up a plus or minus sign. Finally, $T_x$ commutes with the Hamiltonian in the ground state manifold [51]. To show this last property, we first compute the commutator $[\xi_p^+(x_1)\xi_p^-(x_2), H_q]$, where we rewrite the plaquette Hamiltonian as

$$H_p = -\frac{i\nu}{2} \int_{-3\ell}^{3\ell} dx \left( \xi_p^+ \partial_x \xi_p^- + \xi_p \partial_x \xi_p \right). \tag{61}$$

Then it follows that

$$[\xi_p^+(x_1)\xi_p^-(x_2), H_q] = -i\nu \delta_{pq}\left[ \partial_{x_1}\xi_q^+(x_1)\xi_q^-(x_2) + \xi_q^+(x_1)\partial_{x_2}\xi_q^-(x_2) \right], \tag{62}$$

which vanishes when projected onto the plaquette ground state. This can be more easily seen by expanding the right-hand side in Fourier modes, such that we have

$$[\xi_p^+(x_1)\xi_p^-(x_2), H_q] = \delta_{pq} \frac{2\pi\nu}{L} \sum_{k_1 k_2} (k_1 + k_2)\xi_{q,k_1}^+ \xi_{q,k_2}^- e^{i2\pi(k_1 x_1 + k_2 x_2)/L}, \tag{63}$$

with $\xi_{p,k}^\pm = \xi_{p,k}^x \pm i\xi_{p,k}^y$, see Eq. (34). The vacuum expectation value of $\xi_{q,k_1}^+ \xi_{q,k_2}^-$ then enforces $k_1 + k_2 = 0$. Hence, since $T_x$ is just a product of elements of the form $\xi_p^+(x)\xi_p^-(-x)$, and the full Hamiltonian involves the sum over all $H_p$, we conclude the commutator of $T_x$ and $H$ vanishes in the ground state subspace.

Let us now define the operator $T_y$, which transports spin-1 excitations around the vertical direction. We introduce $T_y$ as the product of local operations:

$$T_y \sim \prod_{\langle p_1 p_2 \rangle \in \mathcal{W}_y} \xi_{p_1}^+(-2\ell-\epsilon)\xi_{p_1}^-(-\ell+\epsilon)\xi_{p_2}^+(3\ell-\epsilon)\xi_{p_2}^-(\epsilon), \tag{64}$$

where with $0 < \epsilon < \ell$. We use the notation[1] $\langle p_1 p_2 \rangle$ to indicate that we use a "doubled unit cell" with two types of plaquettes along the closed string $\mathcal{W}_y$, as shown in Fig. 4. We can

---

[1]This notation is only necessary because we have chosen $\mathcal{W}_y$ to run in the direction perpendicular to $\mathcal{W}_x$. Alternatively, we could have defined the closed strings along two independent but nonorthogonal directions, following for instance the standard choice of primitive lattice vectors for the triangular lattice of hexagonal plaquettes.

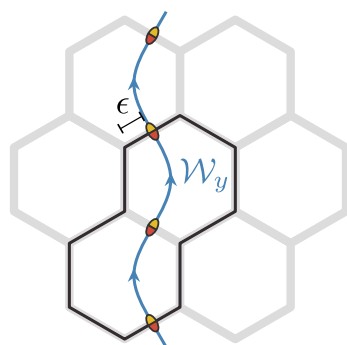

Figure 4: String operator along the vertical direction. The path is parametrized by the distance offset $\epsilon$. The outlined pair of plaquettes corresponds to the double unit cell used in the definition of the closed string $\mathcal{W}_y$.

translate $T_y$ to boson language as well,

$$T_y \sim \prod_{\langle p_1 p_2 \rangle \in \mathcal{W}_y} \exp\left[2i \int_{-\ell+\epsilon}^{-2\ell-\epsilon} dx\, \partial_x \phi_{p_1}(x) + 2i \int_{\epsilon}^{3\ell-\epsilon} dx\, \partial_x \phi_{p_2}(x)\right]. \tag{65}$$

It is easy to verify that the operator $T_y$ enjoys the same properties as $T_x$.

Next, we introduce the operators that realize the transport of the spin-$\frac{1}{2}$ excitations around the torus. The operator that carries the spinon along the horizontal direction is

$$U_x \sim \prod_{p \in \mathcal{W}_x} \mathcal{P}_{\mathbb{1}}\left[u_p^+(\ell+\epsilon) u_p^-(-\ell-\epsilon)\right]\mathcal{P}_{\mathbb{1}}. \tag{66}$$

Unlike the definition of $T_x$, here we need to specify the fusion channel for the twist operators. In boson language, $U_x$ takes the form

$$U_x \sim \prod_{p \in \mathcal{W}_x} e^{i\phi_p(\ell+\epsilon)-i\phi_p(-\ell-\epsilon)} \mathcal{P}_{\mathbb{1}}\left[\sigma_p(\ell+\epsilon)\sigma_p(-\ell-\epsilon)\right]\mathcal{P}_{\mathbb{1}}, \tag{67}$$

where we can take the exponentials out of the projection. Similarly, we define the operator that carries the spinon along the vertical direction as

$$U_y \sim \prod_{\langle p_1 p_2 \rangle \in \mathcal{W}_y} \mathcal{P}_{\mathbb{1}}\left[u_{p_1}^+(-2\ell-\epsilon) u_{p_1}^-(-\ell+\epsilon)\right]\mathcal{P}_{\mathbb{1}}\left[u_{p_2}^+(3\ell-\epsilon) u_{p_2}^-(\epsilon)\right]\mathcal{P}_{\mathbb{1}}. \tag{68}$$

Note that, in analogy with $T_y$, we employ a two-plaquette unit cell in the definition of $U_y$. The bosonized version of $U_y$ reads

$$U_y \sim \prod_{\langle p_1 p_2 \rangle \in \mathcal{W}_y} e^{i\phi_{p_1}(-2\ell-\epsilon)-i\phi_{p_1}(-\ell+\epsilon)+i\phi_{p_2}(3\ell-\epsilon)-i\phi_{p_2}(\epsilon)}$$
$$\times \mathcal{P}_{\mathbb{1}}\left[\sigma_{p_1}(-2\ell-\epsilon)\sigma_{p_1}(-\ell+\epsilon)\right]\mathcal{P}_{\mathbb{1}}\left[\sigma_{p_2}(3\ell-\epsilon)\sigma_{p_2}(\epsilon)\right]\mathcal{P}_{\mathbb{1}}. \tag{69}$$

We now examine the algebra of the string operators, which can be established by the sole use of bosonization (see Appendix A). In effect, we find that the spin-1 operators $T_x$ and $T_y$ commute,

$$T_x T_y = T_y T_x, \tag{70}$$

and the spin-$\frac{1}{2}$ operators satisfy

$$\begin{aligned}
T_x U_x &= U_x T_x, & T_y U_x &= -U_x T_y, \\
T_y U_y &= U_y T_y, & T_x U_y &= -U_y T_x.
\end{aligned} \tag{71}$$

As shown in the appendix, these results are independent of the choice of the $\epsilon$ parameters, which can be taken to be all different for $T_x$, $T_y$, $U_x$, and $U_y$. That is, we are free to choose the position where the strings cross the plaquettes for each operator separately. Equation (70) implies that $T_x$ and $T_y$ form a set of mutually commuting operators. We can then label the ground states by the eigenvalues of $T_x$ and $T_y$, which correspond to the choices of boundary conditions on the torus:

$$|\Omega_{--}\rangle, \qquad |\Omega_{+-}\rangle, \qquad |\Omega_{-+}\rangle, \qquad |\Omega_{++}\rangle. \tag{72}$$

More interestingly, the commutation relations in Eq. (71) imply that the operators $U_x$ and $U_y$ act as global twist operators, changing the boundary conditions on the torus. This means that we can use them to navigate among different ground states of our network. For example, suppose that we start from the vacuum state with antiperiodic boundary conditions in both directions, $|\Omega_{--}\rangle$. We can arrive at the ground states with mixed boundary conditions by acting with $U_x$ and $U_y$ separately, i.e.

$$|\Omega_{-+}\rangle \sim U_x|\Omega_{--}\rangle, \qquad |\Omega_{+-}\rangle \sim U_y|\Omega_{--}\rangle. \tag{73}$$

What about the state $|\Omega_{++}\rangle$? At first sight, it seems that we could reach this state by acting with $U_x U_y$. However, due to the non-Abelian nature of the $SU(2)_2$ anyons, the fourth state is inaccessible from this manifold:

$$|\Omega_{++}\rangle \sim \mathcal{P}_{\text{phys}} U_x U_y |\Omega_{--}\rangle = 0, \tag{74}$$

where $\mathcal{P}_{\text{phys}}$ is a projector onto the physical subspace of the spin-chain network.

To rule out the fourth ground state, let us then take a closer look at the composite operator $U_x U_y$. The nontrivial contribution must come from the plaquette where the two strings meet, i.e. $\mathcal{W}_x \cap \mathcal{W}_y \in p$. Focusing on this plaquette, we have

$$(U_x U_y)_p \sim \mathcal{P}_{\mathbb{1}}\big[\sigma_p(\ell + \epsilon_x)\sigma_p(-\ell - \epsilon_x)\big]\mathcal{P}_{\mathbb{1}}\big[\sigma_p(-2\ell - \epsilon_y)\sigma_p(-\ell + \epsilon_y)\big]\mathcal{P}_{\mathbb{1}}, \tag{75}$$

where we drop the bosonic exponentials because they clearly fuse to the identity. Thus, we just need to show that the twist fields $\sigma$ do not fuse to the identity. Let us then verify that the four-point function vanishes:

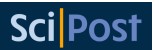 $\langle 0|\mathcal{P}_{\mathbb{1}}(\sigma_1 \sigma_2)\mathcal{P}_{\mathbb{1}}(\sigma_3 \sigma_4)\mathcal{P}_{\mathbb{1}}|0\rangle = 0$, (76)

where $|0\rangle$ denotes the ground state (NS vacuum) of the plaquette. It is important to remark that, as the operators in Eq. (76) are not radial-ordered [42], we are dealing with a nontrivial configuration for the branch cuts, which can be represented by

. (77)

Note that we have reduced our problem to the analysis of a four-point function on a single plaquette. This approach differs significantly from what happens in standard coupled-wire constructions [50], where one does not have such quasilocal degrees of freedom.

Let us quickly recall some basic facts about the braiding rules of the Ising CFT. Our conventions follow Kitaev [7] and Iadecola *et al.* [50]. The fusion space for the four-point function $\langle \sigma_1 \sigma_2 \sigma_3 \sigma_4 \rangle$ is two-dimensional, so we choose the basis

$$|\eta_1\rangle = \quad , \qquad |\eta_2\rangle = \quad . \tag{78}$$

Notice that our diagrams represent a particular channel for the expansion in conformal blocks. This choice is arbitrary. As a matter of fact, we can change the channel expansion using the so-called crossing matrix:

$$= F \quad , \qquad F = \frac{1}{\sqrt{2}} \begin{pmatrix} 1 & 1 \\ 1 & -1 \end{pmatrix}, \tag{79}$$

where we supress the channel index and use the shorthand notation $F \equiv F_\sigma^{\sigma\sigma}$. Another fundamental operation, the exchange of two particles when their fusion channel is held fixed, $\sigma \times \sigma \to \mathcal{O}$ with $\mathcal{O} \in [\mathbb{1}, \xi]$, is implemented by

$$= R_\mathcal{O}^{\sigma\sigma} \quad , \qquad R_\mathcal{O}^{\sigma\sigma} = e^{i\pi(2h_\sigma - h_\mathcal{O})}. \tag{80}$$

Here, $h_\sigma = \frac{1}{16}$ is the conformal dimension of the twist operator. Using that $h_\mathbb{1} = 0$, $h_\xi = \frac{1}{2}$, we define the braiding matrix as

$$= R \quad , \qquad R = \begin{pmatrix} e^{i\pi/8} & 0 \\ 0 & e^{-i3\pi/8} \end{pmatrix}. \tag{81}$$

Note that the matrix is diagonal since the fusion channel is fixed. We finally note that fermion parity is diagonal in the basis we have chosen, with $(-1)^F = \mathcal{P}_\mathbb{1} - \mathcal{P}_\xi$.

We are now ready to resume our discussion of the four-point function. The operator product we are interested in can be associated to

$$\mathcal{P}_\mathbb{1}(\sigma_1 \sigma_2) \mathcal{P}_\mathbb{1}(\sigma_3 \sigma_4) \mathcal{P}_\mathbb{1} \quad \to \quad . \tag{82}$$

Using the rules above, we untie this diagram as

$$\text{(diagram)} = F \, \text{(diagram)} = FR^{-1} \, \text{(diagram)} = FR^{-1}F^{-1} \, \text{(diagram)} = FR^{-1}F^{-1}R \, \text{(diagram)}$$

$$= FR^{-1}F^{-1}RF \, \text{(diagram)} . \tag{83}$$

Our sequence of operations coincides with the ones used in Ref. [50]. Since we start from the identity channel, after we unwind the diagram, we arrive at

$$\text{(diagram)}_{\mathbb{1}} = e^{i\pi/4} \, \text{(diagram)}_{\xi} . \tag{84}$$

We then see that this operation corresponds to changing the fusion channel. The operator product in Eq. (82) is mapped to the radial-ordered product

$$e^{i\pi/4}\mathcal{P}_\xi(\sigma_1\sigma_2)\mathcal{P}_\xi(\sigma_3\sigma_4)\mathcal{P}_\xi , \tag{85}$$

which clearly vanishes when we take the expectation value in the ground state of the plaquette. As we have established before, this condition excludes the fourth state $|\Omega_{++}\rangle$ of the ground-state manifold. We note in passing that this result is associated with a zero component $(S_{\mathbb{1}})_{\sigma\sigma} = 0$ of the topological $S$-matrix [7].

We conclude with a few words about the blocking mechanism in the network. Our solution shows that $U_x U_y$ can only fuse through the fermion channel. However, this possibility is ruled out by the conservation of fermion parity $(-1)^F$. This observation is compatible with a prediction of Read and Green [52], who showed that the ground state of a weak-pairing $p$-wave superconductor has a well-defined fermion parity given the boundary conditions on the torus. As a matter of fact, they found that the three ground states $|\Omega_{--}\rangle$, $|\Omega_{-+}\rangle$, and $|\Omega_{+-}\rangle$ have even fermion number, while $|\Omega_{++}\rangle$ has odd fermion number. The pair wave function of their $p$-wave superconductor resembles the Moore-Read (Pfaffian) quantum Hall state [53]. The connection with our SU(2)$_2$ network model seems natural given that a Pfaffian state was also used by Greiter and Thomale to study a non-Abelian CSL state in $S = 1$ antiferromagnets [9].

## 4  Conclusions

We presented a network construction of topological chiral spin liquids in two spatial dimensions. The construction scheme relies on chiral fixed points of spin-chain junctions. The CSLs inherit their non-Abelian character from the SU(2)$_k$ WZW models that describe the constituent spin chains of the network. We illustrated this approach by looking further into the topological properties of the SU(2)$_2$ model. First, we showed that the theory exhibits emergent Ising anyons. We then constructed a set of string operators that implement transport of elementary excitations around a torus. These operators are used to label and cycle the set of degenerate ground states, demonstrating how the expected threefold degeneracy of the SU(2)$_2$ CSL arises

in the network construction. This calculation makes use of the operator algebra of the underlying CFTs that furnish the low-energy degrees of freedom for the network construction, thus making explicit the connection between these CFTs and the emergent topological phases.

There are a number of open directions for the study of network constructions that are worth exploring further. One interesting possibility is to investigate the even more elusive gapless CSL phases [54–57]. These phases are hard to describe within standard coupled-wire constructions, which hinge on relevant perturbations that gap out the bulk degrees of freedom, but a network with staggered chiral boundary conditions at the junctions may provide a starting point to capture the gapless modes. Another question is whether such a construction could be used to investigate topological phases in three dimensions. Finally, we remark that, while our construction was devised to offer a controllable analytical framework to study CSLs, it may also guide bottom-up approaches to realize these strongly correlated topological phases in artificial quantum materials. This route might involve assembling critical spin chains on a substrate and using circularly polarized light to drive chiral three-spin interactions [58, 59]. Remarkably, recent experiments [60] demonstrated the on-surface synthesis of nanographene $S = 1$ chains with nearly isotropic bilinear and biquadratic exchange interactions that could be tuned to the vicinity of the critical point described by the SU(2)$_2$ WZW model [61–63].

## Acknowledgements

This work was supported by the Brazilian funding agencies CAPES (H.B.X.) and CNPq (R.G.P.), and by DOE Grant No. DE-FG02-06ER46316 (C.C.). Research at IIP-UFRN is funded by Brazilian ministries MEC and MCTI.

## A  Exchange algebra of string operators

In this appendix, we use bosonization to derive the commutation relations in Eqs. (70) and (71). We begin by showing $T_x$ and $T_y$ commute. In the boson representation, see Eqs. (60) and (65), these two operators take the general form

$$T_\alpha \sim \exp[\mathcal{T}_\alpha(\epsilon)], \qquad \alpha \in \{x, y\}, \tag{A.1}$$

where we omit normalization constants, and the operators that appear in the argument of the exponential are

$$\mathcal{T}_x(\epsilon) = 2i \sum_{p \in \mathcal{W}_x} \int_{-\ell-\epsilon}^{\ell+\epsilon} dx\, \partial_x \phi_p,$$

$$\mathcal{T}_y(\epsilon) = 2i \sum_{\langle pp' \rangle \in \mathcal{W}_y} \Big[ \int_{-\ell+\epsilon}^{-2\ell-\epsilon} dx\, \partial_x \phi_p + \int_{\epsilon}^{3\ell-\epsilon} dx\, \partial_x \phi_{p'} \Big]. \tag{A.2}$$

We can obtain the algebra of $T_x$ and $T_y$ by means of the identity $e^A e^B = e^B e^A e^{[A,B]}$, valid for constant $[A, B]$. Let us then compute the commutator of $\mathcal{T}_x$ with $\mathcal{T}_y$. Using the equal-time

algebra of the chiral bosons in Eq. (45), we find

$$
\begin{aligned}
[\mathcal{T}_x(\epsilon), \mathcal{T}_y(\epsilon')] = -i\pi \sum_{p \in \mathcal{W}_x} \sum_{\langle qq' \rangle \in \mathcal{W}_y} \Big\{ &\delta_{pq}\Big[ \mathrm{sgn}(3\ell + \epsilon + \epsilon') - \mathrm{sgn}(2\ell + \epsilon - \epsilon') - \mathrm{sgn}(\ell - \epsilon + \epsilon') \\
&+ \mathrm{sgn}(-\epsilon - \epsilon')\Big] + \delta_{pq'}\Big[ \mathrm{sgn}(-2\ell + \epsilon + \epsilon') - \mathrm{sgn}(\ell + \epsilon - \epsilon') \\
&- \mathrm{sgn}(-4\ell - \epsilon + \epsilon') + \mathrm{sgn}(-\ell - \epsilon - \epsilon')\Big]\Big\}.
\end{aligned} \tag{A.3}
$$

Here we use $\epsilon$ and $\epsilon'$ to denote the free parameters of $T_x$ and $T_y$, respectively. Given that both of them lie in the range $0 < \epsilon, \epsilon' < \ell$, we obtain

$$
[\mathcal{T}_x(\epsilon), \mathcal{T}_y(\epsilon')] = 2\pi i \sum_{p \in \mathcal{W}_x} \sum_{\langle qq' \rangle \in \mathcal{W}_y} \big(\delta_{pq} + \delta_{pq'}\big) = 2\pi i, \tag{A.4}
$$

where the last step follows from the fact that these two strings only meet once on the torus. We thus conclude that the operators $T_x$ and $T_y$ commute, leading us to the relation in Eq. (70). Note that this property is independent of the particular values we may choose for the parameters $\epsilon$ and $\epsilon'$.

Let us now determine the commutation relations among the spin-1 and the spin-$\frac{1}{2}$ operators. We recall that the boson representation of the spin-$\frac{1}{2}$ operators also contains an Ising part, see Eqs. (67) and (69). However, since the Ising and boson components are independent, it is clear that the significant contribution to the exchange relations between $T$ and $U$ operators only come from the bosonized components of both operators, i.e.

$$
T_\alpha U_\beta = U_\beta T_\alpha \exp\big\{[\mathcal{T}_\alpha(\epsilon), \mathcal{U}_\beta(\epsilon')]\big\}, \tag{A.5}
$$

where $\exp[\mathcal{U}_\alpha(\epsilon)]$ denotes the boson part of $U_\alpha$. From Eqs. (67) and (69), we have

$$
\begin{aligned}
\mathcal{U}_x(\epsilon) &= i \sum_{p \in \mathcal{W}_x} \Big[ \phi_p(\ell + \epsilon) - \phi_p(-\ell - \epsilon) \Big], \\
\mathcal{U}_y(\epsilon) &= i \sum_{\langle pp' \rangle \in \mathcal{W}_x} \Big[ \phi_p(-2\ell - \epsilon) - \phi_p(-\ell + \epsilon) + \phi_{p'}(3\ell - \epsilon) - \phi_{p'}(\epsilon) \Big].
\end{aligned} \tag{A.6}
$$

Thus, we just need to evaluate the commutators $[\mathcal{T}_\alpha(\epsilon), \mathcal{U}_\beta(\epsilon')]$ to establish the exchange relations in Eq. (71). The commutator of $\mathcal{T}_x$ with $\mathcal{U}_x$ vanishes identically:

$$
\begin{aligned}
[\mathcal{T}_x(\epsilon), \mathcal{U}_x(\epsilon')] = -\frac{i\pi}{2} \sum_{p \in \mathcal{W}_x} \sum_{q \in \mathcal{W}_x'} \delta_{pq}\Big[ &\mathrm{sgn}(\epsilon - \epsilon') - \mathrm{sgn}(2\ell + \epsilon + \epsilon') \\
&- \mathrm{sgn}(-2\ell - \epsilon - \epsilon') + \mathrm{sgn}(-\epsilon + \epsilon')\Big] = 0.
\end{aligned} \tag{A.7}
$$

Note that this is trivial for the case where we choose two different horizontal paths $\mathcal{W}_x$ and $\mathcal{W}_x'$, with no overlapping plaquettes, but holds true even in the case of superimposed paths, where the contribution of every plaquette cancels out. Hence, $T_x$ and $U_x$ commute, as written in the first relation in Eq. (71).

We then consider the commutator of $\mathcal{T}_y$ with $\mathcal{U}_x$. After using the commutation relations for the chiral bosons in Eq. (45), we arrive at

$$
\begin{aligned}
[\mathcal{T}_y(\epsilon), \mathcal{U}_x(\epsilon')] = -\frac{i\pi}{2} \sum_{\langle pp' \rangle \in \mathcal{W}_y} \sum_{q \in \mathcal{W}_x} \Big\{ &\delta_{pq}\Big[ \mathrm{sgn}(-3\ell - \epsilon - \epsilon') - \mathrm{sgn}(-\ell - \epsilon + \epsilon') \\
&- \mathrm{sgn}(-2\ell + \epsilon - \epsilon') + \mathrm{sgn}(\epsilon + \epsilon')\Big] + \delta_{p'q}\Big[ \mathrm{sgn}(2\ell - \epsilon - \epsilon') - \mathrm{sgn}(4\ell - \epsilon + \epsilon') \\
&- \mathrm{sgn}(-\ell + \epsilon - \epsilon') + \mathrm{sgn}(\ell + \epsilon + \epsilon')\Big]\Big\}.
\end{aligned} \tag{A.8}
$$

Since the parameters $\epsilon$ and $\epsilon'$ are bounded, and the strings only meet at one plaquette, the expression above gives

$$[\mathcal{T}_y(\epsilon), \mathcal{U}_x(\epsilon')] = -i\pi. \tag{A.9}$$

We thus conclude the operators $T_y$ and $U_x$ anticommute, i.e. $T_y U_x = -U_x T_y$. It is straightforward to verify the commutation relations of $\mathcal{U}_y$ as well. The commutators of interest are

$$[\mathcal{T}_x(\epsilon), \mathcal{U}_y(\epsilon')] = i\pi, \qquad [\mathcal{T}_y(\epsilon), \mathcal{U}_y(\epsilon')] = 0. \tag{A.10}$$

From these it follows that $U_y$ commutes with $T_y$, but anticommutes with $T_x$, completing the set of exchange relations in Eq. (71).

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
