# Peer review of "Network construction of non-Abelian chiral spin liquids"

_SciPost Physics, doi:SciPost Phys. 14, 105 (2023)_

## Round 1 · Referee Report · Anonymous (Referee 1) · 2022-12-20

Strengths

1- A new theoretical approach to the description of non-Abelian topological phases in two spatial dimensions starting with 1D spin chains. It is an alternative to coupled-wire constructions and in this respect it could be very useful to find new physical realisations of exotic phases of matter. 2- The paper is very well written, accessible for most readers in the field.

Weaknesses

1- The main weakness is that the authors started the approach by postulating the existence of a chiral fixed point for the junction for general values of k. For k>2 it requires many fine tunings to kill unwanted relevant and marginal perturbations. 2- The analysis is restricted to k=2 and exploits the free-field representation of SU(2)_2 CFT in terms a bosonic field and the Ising CFT. It will be interesting to see wether the explicit construction of string operators that implement the transport of quasiparticles around the torus could also be done in the k>2 case.

Report

In this paper, the authors have introduced a network construction of non
Abelian SU(2)_2 chiral spin liquid in two dimensions which exhibits Ising
anyons. Such an exotic phase has already been described in the past by
means of different coupled wires constructions. However, the latter
approach requires several fine tunings and specific interactions which are
difficult to realize in quantum materials. One interest of this paper is to
describe this phase with a network of chiral Y junctions on a honeycomb
lattice. The main assumption is that each Y junction is fine tuned to a chiral fixed point which is described by an SU(2)_k conformal field theory (CFT).
For k=2, the case of interest here, it requires the fine tuning of a single
relevant operator and a marginal one. The topological phase of the
non-Abelian chiral spin liquid phase are explicitly shown by exploiting
a free field representation of the SU(2)_2 CFT. In particular, they found
the correct edge states, described by a chiral SU(2)_2 CFT, the existence of a bulk gap, which is guaranteed by
the construction using a collection of a finite critical spin chains, the
description of the low-lying excitations, and the correct topological
degeneracy on the torus for Ising anyons.

The paper is very interesting and very well written. It provides a new
controllable analytical approach to the description of exotic topological
phases in two dimensions which could be useful to find
experimental realization of these phases in the context of artificial
materials. I strongly recommend the publication of this paper in SciPost
journal.

Requested changes

1- A very minor remark is the citation of the paper of Capponi et al., Phys. Rev. B 88, 075132 (2013), which found an SU(2)_k quantum criticality in ladder systems with ring-exchange interactions for all k > 1. For completeness, this paper could be cited together with the series of papers [35-39] on critical spin chains.

---

## Round 1 · Referee Report · Anonymous (Referee 2) · 2023-1-11

Strengths

  1. The paper presents a view of a non-abelian 2d topological phase constructed from 1d critical models, which are extremely well studied and understood. This makes the former accessible to people familiar with the latter.
  2. The paper is written quite clearly and even pedagogically.

Weaknesses

  1. The universal content of the described topological phase is well-known. What this paper does is reproduce it, rather than obtain new conclusions.
  2. As a construction, it has the defect that the chiral boundary condition is put in by hand. This is only a weak defect that does not affect the goals of the paper, and indeed makes readability easier. However it means that the model does not a priori appear especially realistic.

Report

I do recommend acceptance. The paper satisfies criterion 4 to "provide a novel and synergistic link between research areas", in this case 2d topological phases and 1d critical models. I also think that there is potential for follow-up work (criteria 3) for example to 2d critical theories.

---

## Editorial Decision

published